# An Active Image-Based Mobile Food Record Is Feasible for Capturing Eating Occasions among Infants Ages 3–12 Months Old in Hawai‘i

**DOI:** 10.3390/nu14051075

**Published:** 2022-03-03

**Authors:** Marie K. Fialkowski, Jessie Kai, Christina Young, Gemady Langfelder, Jacqueline Ng-Osorio, Zeman Shao, Fengqing Zhu, Deborah A. Kerr, Carol J. Boushey

**Affiliations:** 1Department of Human Nutrition Food and Animal Sciences, College of Tropical Agriculture and Human Resources, University of Hawaiʻi at Mānoa, 1955 East West Road, Agricultural Sciences 216, Honolulu, HI 96822, USA; jessiek@hawaii.edu (J.K.); chyoung3@hawaii.edu (C.Y.); glangfel@hawaii.edu (G.L.); 2Department of Psychiatry, John A. Burns School of Medicine, University of Hawaiʻi at Mānoa, 677 Ala Moana Blvd., Honolulu, HI 96813, USA; ng-osorioj@dop.hawaii.edu; 3School of Electrical and Computer Engineering, Purdue University, 610 Purdue Mall, West Lafayette, IN 47907, USA; shao112@purdue.edu (Z.S.); zhu0@purdue.edu (F.Z.); 4Curtin School of Population Health, Curtin University, Kent Street, Bentley Western Australia, Bentley, WA 6102, Australia; d.kerr@curtin.edu.au; 5University of Hawaiʻi Cancer Center, 701 Ilalo St, Honolulu, HI 96813, USA; cjboushey@cc.hawaii.edu

**Keywords:** infant diet, mobile food record technology, acceptability

## Abstract

The ability to comprehensively assess the diet of infants is essential for monitoring adequate growth; however, it is challenging to assess dietary intake with a high level of accuracy. Infants rely on surrogate reporting by caregivers. This study aimed to determine if surrogate reporters (e.g., caregivers) could use an image-based mobile food record adapted (baby mFR) to record infants’ eating occasions, and via caregiver feedback, could assess the usability and feasibility of the baby mFR in recording infants’ diets. This was a cross-sectional study in which surrogate reporters (e.g., caregivers) recorded all food and beverage intake (including human milk) of the infant over a 4-day period. Trained research staff evaluated all images submitted during data collection for different indicators of quality. All surrogate reporters were asked to complete a usability questionnaire at the end of the 4-day data collection period. Basic descriptive analyses were performed on the infants 3–12 months of age (*n* = 70). A total of 91% (*n* = 64) of surrogate reporters used the baby mFR to record their infants’ eating occasions. The mean number of images submitted daily per participant via the mFR was 4.2 (SD 0.2). A majority of submitted images contained the fiducial marker and the food and/or beverage was completely visible. The mFR was found to be easy to use; however, suggestions were provided to increase utility of the application such as the inclusion of a bottle button and reminders. An image-based dietary assessment method using a mobile app was found to be feasible for surrogate reporters to record an infant’s food and beverage intake throughout the day.

## 1. Introduction

Adequate nutrition is important for achieving normal growth and development during infancy. Therefore, dietary assessment is an integral component of infant growth monitoring [1]. However, the accuracy of the assessment of complementary feeding and dietary intakes in infants is reliant on surrogate reporting by caregivers. Surrogate reporting is influenced by the amount of time spent with the infant and knowing the full details of any food prepared elsewhere (e.g., at daycare) during the assessment period. A systematic review concluded weighed food records provided the best estimates of energy as estimated by doubly labeled water for children 0.5–4 years of age [2]. This method, however, places a much higher burden on those undertaking the recording (aka infant surrogate reporters).

Mobile technology offers a wide range of feasible options for dietary assessment, which are easier to incorporate into daily routines due to its widespread adoption. Incorporating user images captured by mobile technology has improved the accuracy of conventional dietary assessment methods by adding eating occasion details such as time as well as reducing underreporting when compared with traditional assessment methods [3].

One image-based method, the Technology Assisted Dietary Assessment (TADA) system [4,5], uses an image-based dietary record application, the mobile food record (mFR) [6,7], to capture before and after eating images from the mobile device. Automated image analysis [8] or a trained analyst can identify the foods in the image and estimate portion size of foods consumed [9,10,11]. This method provides real-time data collection and eliminates reliance on the respondent’s memory, proxy reports, and ability to write and/or estimate portions [9]. The mFR has been identified as feasible for use in assessing the diets of children as young as 3 years [10] of age and through to adults [12,13,14]. Adults (18–49 y) are the highest adopters of mobile devices and represent the majority of parents with young children [15]. Therefore, determining the feasibility, or ease and convenience of use, by surrogates (aka caregivers) of the mFR for infant dietary assessment merits exploration. This study is the first time that an image-based dietary assessment approach has been used to capture the total diet of infants which may improve accuracy of dietary assessment in infants through reducing surrogate burden [3]. Addressing feasibility, which includes determining willingness, adherence and compliance, are all an important aspects to evaluate as they inform important parameters for further research such as validation studies [16]. The purpose of this study was to determine if surrogate reporters (e.g., caregivers) could successfully use the mFR adapted for infants, referred to as the baby mFR in this paper, to record their infants’ eating occasions over a 4-day period. In addition, this study gathered the perspective of the caregivers on usability of the baby mFR. This study addresses identified recommendations for using a mobile app specifically designed for capturing dietary assessment of infants [17].

## 2. Materials and Methods

This cross-sectional examination of the baby mFR was conducted with an ethnically diverse sample of infants between 3 and 12 months of age residing on O‘ahu, Hawai‘i between March 2018 and February 2019. This age range was selected in order to examine complementary food introduction and diet diversity, which was reported in a separate publication [18].

Eligible participants in this study were infants between 3 and 12 months of age that had already started complementary feeding, infants identified by their surrogate reporters as having Native Hawaiian, Pacific Islander, and/or Filipino ethnicity, and infants who had a surrogate reporter that was 18 years of age or older. Additional eligibility for surrogate reporters included access to an iOS mobile device with reliable access to the Internet. This sample of convenience was primarily recruited through community events (e.g., Baby Expo), community programs (e.g., Special Supplemental Nutrition Program for Women, Infants, and Children), and social networking (e.g., Facebook groups). This study was deemed Exempt by the University of Hawai‘i Institutional Review Board. Consent was obtained in writing from the surrogate reporters for both their participation and their infant’s participation prior to collecting any data. Surrogate reporters were compensated with a USD 40 gift card for their participation.

Surrogate reporting via caregivers using the baby mFR provided the dietary data to do the assessment. The baby mFR collected breastfeeding occasion via a timer feature and complementary feeding occasions via a camera feature capturing pre- and post-images of all foods and beverages consumed.

The baby mFR is a mobile application running on an iOS platform designed to use the camera on a smart device to capture food and beverage intake, which is then used to estimate energy and nutrient intakes. All captured images were automatically uploaded to the secure TADA website, when a wireless connection was available. Each surrogate was also provided with fiducial markers (FM) to include in all recorded images in the bottom left corner. The FM acts as a size and color reference which enhances the ability of the trained analyst to assess the content of the image or for automated analysis to occur as part of an image recognition system [19]. Prior to data collection, the baby mFR app was loaded on to the surrogate reporter’s mobile device and the surrogate reporter was trained on its use. The data collection period was informed by previous studies [10,20] as this was the first time this method was applied to infants. Surrogate reporters were instructed to complete a 4-day food record (Thursday–Sunday) using the baby mFR. Pre- and post-images of all foods and beverages consumed by the infant participant over the 4-day period were captured by the baby mFR. The baby mFR included interchangeable color borders (i.e., red or green) to guide the surrogate on when to take the image as published elsewhere [7,10]. At the conclusion of the 4-day collection period, researchers logged into the TADA website to view images [7]. As needed, images from the baby mFR were reviewed with the surrogate reporters by a member of the research team to verify content and to probe for any forgotten foods or beverages. A unique feature of the baby mFR was a timer to record the start and end of a breastfeeding event. See Figure 1 for a depiction of the baby mFR home screen.

All surrogate reporters were asked to complete an online questionnaire administered through a research web application. Surrogates reported basic demographic information of the infant participant (e.g., age, race/ethnicity) and answered a semi-qualitative usability questionnaire modified from a previous study in children [10]. The questions included in the usability questionnaire are presented in Appendix A.

For this study a human analyst examined all images on the TADA website. Information about each image was entered into a Google Form specifically designed for this study. Such information included the date and timestamp of each image, presence of before or after images, visibility of all foods and beverages and/or the FM, and other non-food objects captured in the image. Breastfeeding timing data and surrogate reporter’s responses to the baby mFR usability questionnaire were also summarized. Open-ended responses were used to provide further context to Likert responses in the usability questionnaire (See Appendix A for examples). All quantitative data were analyzed descriptively (frequencies and means) using IBM SPSS statistics version 27 (IBM Corporation, Armonk, NY, USA).

## 3. Results

### 3.1. Participants

A total of 70 infants and their surrogate reporters participated in the study. As indicated in Table 1, a majority of the infants were between the ages of 6 and 12 months, and were identified by their caregiver as Native Hawaiian or Part Native Hawaiian ethnicity. The study was almost evenly distributed between boys and girls, and those who were given and not given human milk (Table 1). 

### 3.2. Mobile Food Record

The majority of participants (*n* = 64, 91%) recorded 4 out of 4 days of dietary intake. Three participants provided 3 days of dietary intake (4%), two participant completed 2 days of dietary intake (3%), and one participant did 1 day (1%). A total of 66 surrogate reporters (94%) used the baby mFR app to record their infant’s food and beverage intake. Four surrogate reporters (6%) used their mobile device to take images of their infant’s food and beverage intake and texted these images directly to the researchers. Twenty-three (33%) surrogate reporters used both the baby mFR app to upload images and texted food/beverage images or information to the researchers. The mean number of images submitted per day per participant via the mFR app was 4.2 (SD 0.2). As described in Table 2, over 50% of before and after images had a FM. The food was completely visible in almost 75% of the before (*n* = 414) and after images (*n* = 412).

Over 40% (*n* = 30) of all the infant participants in this study had breastfeeding timing data. The mean number of breastfeeding events each day were similar; however, the mean duration of a breastfeeding event in minutes was higher during the weekend days (Saturday and Sunday) versus the weekdays (Thursday and Friday). A majority of breastfeeding events were between 1 min and 120 min in duration (Table 3).

### 3.3. Surrogate Reporter Feedback

When surrogate reporters were asked how long they would be willing to record their baby’s intake using the baby mFR, 35% (*n* = 25) responded for seven days and 34% (*n* = 24) responded for four days. A majority of surrogate reporters (*n* = 64, 91%) indicated the baby mFR did not change the way they were feeding their baby. Eighty percent (*n* = 56) of surrogate reporters indicated they never or almost never had problems using the mFR. Of those that indicated they had problems the most common issues were related to the speed of uploading images, recognition of the FM, image clarity, and the breastfeeding timer.

As shown in Table 4, a majority of surrogate reporters found the mFR easy to use. Reasons provided by surrogate reporters on its ease were related to the convenience of the app being on their phone, minimalist design, and clearly labeled buttons. “It was easy to use and since I carry my phone with me wherever I go, it was much easier to use than writing everything down”, “Because the button selections were minimal, it made it very dummy proof and easy to use”, “The app was clearly labeled and I liked the marker used to determine the angle of each picture. It was user friendly and the ease of it made it easier for me to access and use it properly”, and “They [mFR icons] were relatable. I knew what they were indicating”. A majority of surrogate reporters also found taking before and after eating images, as well as including the FM in the images easy to do. One surrogate reporter commented “Kept one fiducial market in my purse for on the go and one in the kitchen”.

One area in which surrogate reporters found to be not easy to do was starting and stopping the breastfeeding timer (Table 4). Reasons shared by surrogate reporters included “The timer went off when I switched to another app” and “Because I breastfeed whenever, wherever, it was difficult for me to keep up with recording my feedings because I don’t always have my phone by me”.

Caregivers overwhelmingly agreed knowing the purpose of the study motivated their use (Table 4), commenting “I liked knowing I was helping research concerning my child” and “I liked taking pics of my daughter’s food and knowing I was helping you folks out”.

Other notable comments provided by surrogate reporters included the importance of other individuals having the mFR downloaded to their devices to record their baby’s intake. Comments from surrogate reporters included “Actually, my mom had to because I had to go to work 3 out of the 4 days of the study. So she downloaded it to her phone and took most of the lunch and dinner pictures for me” and “Boss helped take pics, she was ok, and boyfriend also used it”. In addition, surrogate reporters’ provided suggestions for improvements to the mFR. Improvements were related to the icons used such as, “A simple example of before and after pictures would have been better instead of three babies making it too busy icons”, as well as for other features “I would suggest having a bottle option on the menu to select the number of ounces instead of needing to take a picture of it”, “Would be nice to add texts and self-identify what it is”, and “more reminders”.

## 4. Discussion

This is the first study to evaluate the use of an image-based novel dietary assessment application, the baby mFR, to capture the diet of Native Hawaiian, Pacific Islander, and/or Filipino infants 3–12 months of age, a population underrepresented in research [21]. These results indicate that the mFR is a feasible method to conduct dietary assessment in infants. The baby mFR provides great utility to infant dietary assessment as it reduces the burden of other traditional methods of dietary assessment such as a dietary record and recall. The use of mobile technology, which is widely prevalent [15], provides data in real time without the burden of completing hand-written dietary records. As demonstrated in this study’s findings, a majority of surrogate reporters were willing to use the mFR in another study indicating a positive reception to the application.

An unexpected outcome of this study, which was only seen in one other study which used the mFR in young adults with Down Syndrome [14] but not seen in previous mFR studies with children or adults [9,10], was the submission of images directly from the surrogate reporters’ mobile device via text message versus through the baby mFR. As indicated in the feedback from surrogate reporters, providing dietary intake information on their infant was a communal affair where other family members, friends, babysitter, or other acquaintances would contribute images on the infants’ intake when the surrogate reporter was not present. Since the baby mFR was only installed on the primary surrogate reporter’s device, images taken from other devices were submitted via text messages to the research team. Future research is needed to explore the utility of uploading the baby mFR on multiple devices for conducting infant dietary assessment. This potentially would also ameliorate the number of eating occasions missing image(s) due to the infant being cared for by someone without access to the baby mFR.

The baby mFR was developed to assess at-the-breast feeding sessions through the use of a timer, which would be less burdensome and costly than other measures of assessment [22]. This feature allowed researchers to distinguish between human milk received directly from the breast versus expressed [23]. Distinguishing how human milk is received by infants may be important as a previous study has shown that health outcomes differ between the feeding modes [24]. Further work is needed to evaluate the breastfeeding timing data for outliers. In this paper, three different cut-offs were used, <1 min, between 1 and 120 min and >120 min. There are various ways in which timing data can be interpreted such as less than 5 [17] or 10 [25] minutes as snacks or lasting as long as 120 min [26]. However, duration of the at-the-breast sessions may have significant variability such as night feeds lasting longer than during the day feeds, and differences in the length of feeds for younger versus older infants. In addition, there seemed to be a higher number of short timing sessions, e.g., <1 min, in the first day of data collection than in later days. This may be a result of reporters getting familiar with the feature’s functionality. However, the frequency of breastfeeding events found in this study were similar to what has been documented elsewhere [27]. Another consideration to increase functionality of the mFR for infant dietary assessment, as suggested by participant caregivers, would be to include a bottle feature which would allow researchers to more easily assess expressed milk, formula, and other liquids provided to infants.

As this study was limited to only iOS users, future studies should explore the mFR use on other mobile device operating systems. Additional studies are needed to validate the mFR such as with biomarkers or a controlled feeding trial to assess reporting accuracy, especially in regard to estimating the volume of human milk consumed during breastfeeding events which are difficult to obtain precisely [28]. Further validation of the mFR to perform automated image analysis including food identification, portion size and contextual processing within the infant population would also be warranted similarly to what has been done among adults and adolescents [11,29,30,31]. Accurate infant dietary assessment data are especially important to inform future iterations of the recently added U.S. Dietary Guidelines for children 0–24 months [32,33].

## 5. Conclusions

An active image-based dietary assessment method using a mobile device was found to be feasible, convenient, and easy to use for caregivers’ to record their infant’s food and beverage intake throughout the day. The significance of this study is that it is the first to evaluate the surrogate use of the mFR among infants younger than 12 months and warrants further research validating its accuracy.

## Figures and Tables

**Figure 1 nutrients-14-01075-f001:**
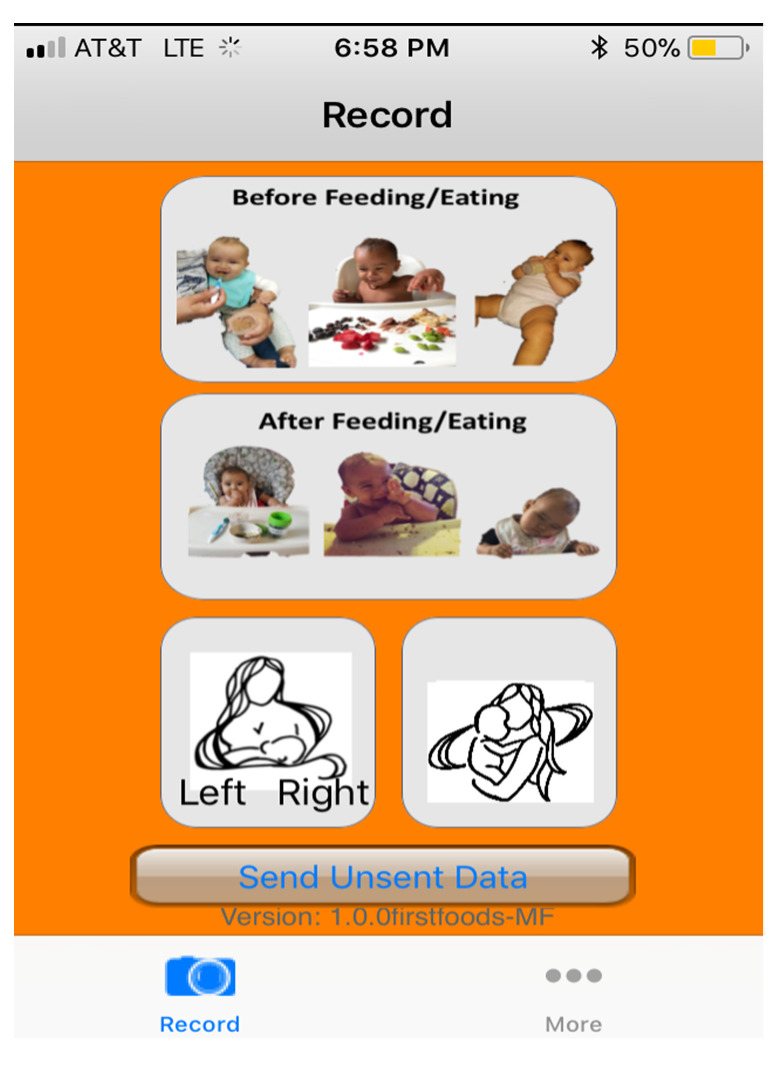
Home screen for the baby Mobile Food Record (mFR) application.

**Table 1 nutrients-14-01075-t001:** Characteristics of infants, 3–12 months of age, that participated in the study (*n* = 70).

Characteristics	*n* (%) ^a^
Age group	
3–5 months	14 (20)
6–12 months	56 (80)
Sex	
Boy	38 (54)
Girl	32 (46)
Ethnicity ^b^	
Part-Native Hawaiian or Native Hawaiian	50 (71)
Pacific Islander only ^c^	4 (6)
Part-Filipino or Filipino	35 (50)
Currently receiving human milk	40 (57)

^a^ Percentages may not add up to 100 due to rounding, ^b^ More than one race/ethnicity may have been self-selected, therefore will not add to 100%, ^c^ Self-reported Pacific Islander ethnic groups included Chamorro, Samoan, Tongan, Maori, Tahitian, and Micronesian.

**Table 2 nutrients-14-01075-t002:** Assessment of images captured using the baby mobile food record (mFR) or via mobile device among infants, 3–12 months of age (*n* = 66).

	Mode of Submission			
	mFR		Text	
Total Number of Images	1114		66	
	Before	After	Before	After
Fiducial Marker	*n* (%) ^a^	*n* (%) ^a^	*n* (%) ^a^	*n* (%) ^a^
Absent	165 (30)	176 (32)	42 (100)	24 (100)
Partially Present	65 (12)	60 (11)	0 (0)	0 (0)
Present	329 (59)	319 (58)	0 (0)	0 (0)
Location of Fiducial Marker in Image				
Absent	165 (30)	176 (32)	42 (100)	24 (100)
Bottom-Left Corner	198 (35)	185 (33)	0 (0)	0 (0)
Bottom-Right Corner	79 (14)	70 (13)	0 (0)	0 (0)
Center Bottom	97 (17)	101 (18)	0 (0)	0 (0)
Center Top	4 (1)	6 (1)	0 (0)	0 (0)
Top-Left Corner	9 (2)	9 (2)	0 (0)	0 (0)
Top-Right Corner	7 (1)	8 (1)	0 (0)	0 (0)
Food and Beverage Visibility				
Forgot to Take an Image of Eating Occasion	8 (1)	36 (7)	0 (0)	0 (0)
Completely Visible	414 (74)	412 (74)	36 (86)	20 (83)
Partially Visible	118 (21)	99 (18)	5 (12)	4 (17)
Not Visible	19 (3)	8 (1)	1 (2)	0 (0)
Participant In Image	30 (5)	37 (7)	5 (12)	2 (8)

^a^ Disclaimer: may not add up to 100 due to rounding.

**Table 3 nutrients-14-01075-t003:** Descriptive data of breastfeeding events of infants, 3–12 months of age (*n* = 30), captured using the mobile food record (mFR) breastfeeding timer.

	mFR Recording Day			
Variable	Thursday	Friday	Saturday	Sunday
Sample size (*n*)	25	30	27	25
	Mean (SD)	Mean SD	Mean SD	Mean SD
Number of breastfeeding events recorded	4.9 (3.5)	5 (3.6)	5.6 (3.4)	4.1 (3.1)
Duration of breastfeeding events in minutes	7.0 (6.3)	12.8 (11.8)	17.5 (18.3)	36.1 (48.4)
	*n* (%)	*n* (%)	*n* (%)	*n* (%)
Number of breastfeeding events <1 min	20 (16)	6 (4)	18 (12)	4 (4)
Number of breastfeeding events between 1 and 120 min	102 (84)	143 (95)	132 (87)	92 (89)
Number of breastfeeding events >120 min	0 (0)	1 (1)	2 (1)	7 (7)

**Table 4 nutrients-14-01075-t004:** Likert responses from caregivers of infants 3–12 months of age (*n* = 70) to the baby mobile food record (mFR) usability questionnaire.

	Strongly Agree or Agree	Neither Agree or Disagree	Disagree or Strongly Disagree
Questions	*n* (%) ^a^	*n* (%) ^a^	*n* (%) ^a^
The mFR was easy to use	66 (94)	3 (4)	1 (1)
The directions about how to use the mFR were easy to follow	65 (93)	3 (4)	2 (3)
Knowing when to take an image of my child’s eating was easy	65 (93)	4 (6)	1 (1)
Remembering to take an image before my child ate was easy	50 (71)	10 (14)	10 (14)
Remembering to take an image after my child ate was easy	45 (64)	15 (21)	10 (14)
Remembering to push the button before breastfeeding my child was easy ^b^	17 (25)	10 (14)	12 (17)
Remembering to push the button after breastfeeding my child was easy ^b^	11 (15)	9 (13)	19 (27)
I found it easy to include the fiducial marker in the picture of my child’s meals	42 (60)	13 (19)	15 (22)
The mFR interfered with my daily activities	2 (3)	19 (27)	49 (70)
Understanding the purpose of the mFR motivated me to use it ^c^	61 (87)	7 (10)	1 (1)
Overall, the mFR was a nuisance to use	3 (4)	19 (27)	48 (69)
Overall, the mFR was enjoyable to use	53 (76)	16 (23)	1 (1)
I would like to participate in another study using the mobile food record.	58 (83)	10 (14)	2 (2)

^a^ Not all percentages will add up to 100 due to rounding. ^b^ This question was not applicable to all participants. ^c^ There was one missing response.

## Data Availability

The data presented in this study are available on request from the corresponding author. The data are not publicly available due to ethical restrictions.

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
