# Peer review of "An Active Image-Based Mobile Food Record Is Feasible for Capturing Eating Occasions among Infants Ages 3–12 Months Old in Hawai‘i"

_nutrients, 2022, doi:10.3390/nu14051075_

Round 1

Reviewer 1 Report

This is a potentially interesting and valuable paper but you have not demonstrated this. It is a descriptive account of a data collection system, the Active Image-Based Mobile Food Record, and an assessment of its useability and acceptability for users. It would be useful if experience from this study, and, perhaps previous evaluations of the tool, could be included to demonstrate how the mFR is calibrated and validated. It would be reassuring if your information could be reframed to demonstrate the intrinsic credibility of the process of collecting and analysing the data. I am presuming that the Fiduciary Markers are technical QC measures, and that they do not aid the quantitative and qualitative evaluation of the infant's dietary intakes. It would be an asset if you could provide this information;- particularly given the variable use of the FM. Would the value of the tool be enhanced if the use of the FM were standardised? The last point seems to apply also to the images of the foods and beverages.

The value and generalised applicability of this paper would be enhanced if the experiential details could be seen to be sounder than just "feasible"

Reviewer 2 Report

The publication “An Active Image-Based Mobile Food Record is Feasible for Capturing Eating Occasions Among Infants Ages 3 - 12 Months  Old in HawaiÊ»i”   by  Marie K. Fialkowski and coworkers evaluates the feasibility, ease and convenience of use by caregivers the image-based mobile food record adapted (baby mFR) among infants younger than 12 months.

The topic of the study is not particularly original, but results are properly reported and adequately summarized in 4 tables. The study is informative and clinically significant.

Reviewer 3 Report

The manuscript concerns the usability and feasibility of the use of an image-based food record application to capture the diet of infants 3 – 12 months of age. I read the work very carefully. It’s well written.

Unfortunately, in my opinion, its major disadvantage is the lack of nutritional data and that there are no comparisons with other tools already available. Would it be possible to do this analysis? If not, I would suggest adding this limit in the discussion

Round 2

Reviewer 1 Report

I have nothing further to offer.